# GC-MS Metabolomics and Antifungal Characteristics of Autochthonous *Lactobacillus* Strains

Paola Scano [1,*], M. Barbara Pisano [2], Antonio Murgia [1], Sofia Cosentino [2] and Pierluigi Caboni [1]

1. Department of Life and Environmental Sciences, University of Cagliari, SS 554 km 4.5, 09042 Monserrato, Italy; a-murgia@hotmail.it (A.M.); caboni@unica.it (P.C.)
2. Department of Medical Sciences and Public Health, University of Cagliari, SS 554 km 4.5, 09042 Monserrato, Italy; barbara.pisano@unica.it (M.B.P.); scosenti@unica.it (S.C.)
* Correspondence: scano@unica.it; Tel.: +39-070-675-4391

**Abstract:** *Lactobacillus* strains with the potential of protecting fresh dairy products from spoilage were studied. Metabolism and antifungal activity of different *L. plantarum*, *L. brevis*, and *L. sakei* strains, isolated from Sardinian dairy and meat products, were assessed. The metabolite fingerprint of each strain was obtained by GC-MS and data submitted to multivariate statistical analysis. The discriminant analysis correctly classified samples to the *Lactobacillus* species and indicated that, with respect to the other species, *L. plantarum* had higher levels of organic acids, while *L. brevis* and *L. sakei* showed higher levels of sugars than *L. plantarum*. Partial Least Square (PLS) regression correlated the GC-MS metabolites to the antifungal activity ($p < 0.05$) of *Lactobacillus* strains and indicated that organic acids and oleamide are positively related with this ability. Some of the metabolites identified in this study have been reported to possess health promoting proprieties. These overall results suggest that the GC-MS-based metabolomic approach is a useful tool for the characterization of *Lactobacillus* strains as biopreservatives.

**Keywords:** *Lactobacillus*; antifungal activity; fresh cheese; biopreservation; GC-MS; metabolomics

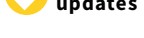

## 1. Introduction

In the Mediterranean area, to revive the ovine dairy industry, the development of new fresh dairy products is increasing. Taking into account that the physico-chemical characteristics of these products greatly favor the growth of pathogenic and spoilage microbes, many research activities have been devoted to finding solutions to protect fresh dairy products from deterioration and to prolong their shelf life while preserving the organoleptic and nutritional properties. Furthermore, in recent years, the increased request for natural foods has challenged the food producers and researchers to find natural alternatives to synthetic preservatives. Among natural food preservation techniques, bio-preservation has gained particular attention; this term refers to the use of microbial strains or metabolites able to inhibit the growth of spoilage and pathogenic microbes in foods and thus improving safety and extending their shelf-life [1]. Lactic acid bacteria (LAB), naturally present in many fermented and not fermented foods of animal and vegetable origin, are considered good candidates as bio-preservative, as they possess numerous functional properties related to the improvement of food safety and health benefits [2,3]. Among LAB, several strains belonging to the genus *Lactobacillus*, frequently present in milk and cheese as the prevalent nonstarter LAB, have shown distinct antimicrobial and antifungal activities [4]. Additionally, *Lactobacillus sakei* strains, characteristic of meat products, have shown antifungal activity [4]. The latter strains can be found in raw meat products stored under vacuum and refrigerated, as well as in fermented sausages, due to their metabolic properties and phenotypic features that are particularly well adapted to growth and survival under the conditions found during meat processing and storage [5,6].

The antifungal activity of LAB may be related to the synergistic action of several compounds, e.g., organic acids (acetic acid, lactic acid, propionic acid, and phenyllactic acid), hydrogen peroxide, cyclic dipeptides, proteinaceous compounds (bacteriocins), and fatty acids [2,7], and it is recognized that the capacity to synthetize these compounds is a strain-linked trait [4]. In our previous work, we found that selected *Lactobacillus* strains of food origin could control mold contamination without altering the sensory characteristics in miniature Caciotta ovine cheese, produced in a laboratory scale [4]; one limitation of this study is that the compounds responsible for the antifungal activity were not studied. Metabolomics is one of the most valuable techniques for the study of metabolite profiles in food matrices [8], and gas chromatography coupled with mass spectrometry (GC-MS) is a well-suited analytical platform for the study of microbial metabolites [9,10]. By the combined approach of GC-MS and multivariate statistical data analysis it was possible to find links between metabolites of dairy products and their microbial profiles [11]. The aim of this work was to test the suitability of this approach for the identification of the metabolites involved in the bioprotective characteristics of autochthonous *Lactobacillus* strains isolated from Sardinian dairy and meat products and to seek correlations between metabolites (or group of metabolites) and antifungal activity. To this goal, we studied the polar and semi-polar low molecular weight metabolites present in the cultured broth of different *Lactobacillus* strains. Antifungal activity of these strains against mold species, commonly occurring in cheese spoilage, was also assessed and results correlated to the GC-MS data by multivariate regression analysis.

## 2. Materials and Methods

### 2.1. Microorganisms and Cultivation Conditions

A total of 9 autochthonous *Lactobacillus* strains (3 *Lactobacillus plantarum*, 2 *Lactobacillus brevis*, and 4 *Lactobacillus sakei*), deposited in the MBDS culture collection (www.mbds.it, accessed on 25 May 2021) were studied. They were isolated from ovine raw milk and artisanal cheeses (*L. plantarum* and *L. brevis*), and sausages (*L. sakei*) manufactured in Sardinia (Table 1), and they were classified on the basis of phenotypic characteristics and molecular methods through polymerase chain reaction amplification. *Lactobacillus* strains were stored at −20 °C in DeMan Rogosa Sharpe (MRS) broth (Microbiol, Cagliari, Italy) with 15% (*v/v*) glycerol and routinely cultivated on MRS agar plates for 48 h at 30 °C in microaerophylia. Each strain was subcultered twice in MRS broth prior to experimental use. Carbohydrate fermentation profiles of the *Lactobacillus* strains was assessed and reported in Table S1.

**Table 1.** List of the *Lactobacillus* strains and food origin.

| Species | Strains | Origin | Molecular Identification | MBDS # |
|---|---|---|---|---|
| *Lactobacillus plantarum* | 4/16898 | Raw sheep's milk | Species- specific PCR [12] | UNICAB27 |
| *Lactobacillus plantarum* | 1/14537 | Raw sheep's milk | Species- specific PCR [12] | UNICAB32 |
| *Lactobacillus plantarum* | C1/15 | Sheep's cheese | Species- specific PCR [12] | UNICAB212 |
| *Lactobacillus brevis* | DSM 32516 | Sheep's cheese | 16S rRNA gene sequencing Universal primers F357-R518 [13] | UNICAB24 |
| *Lactobacillus brevis* | M8/1 | Sheep's cheese | 16S rRNA gene sequencing Universal primers F357-R518 [13] | UNICAB456 |
| *Lactobacillus sakei* | S3 | Artisanal sausage | 16S rRNA gene sequencing Universal primers Y1-Y2 [14] | UNICAB457 |
| *Lactobacillus sakei* | S5 | Artisanal sausage | 16S rRNA gene sequencing Universal primers Y1-Y2 [14] | UNICAB458 |
| *Lactobacillus sakei* | S4 | Artisanal sausage | 16S rRNA gene sequencing Universal primers F357-R518 [13] | UNICAB459 |
| *Lactobacillus sakei* | S3/1 | Artisanal sausage | 16S rRNA gene sequencing Universal primers Y1-Y2 [14] | UNICAB460 |

## 2.2. Extraction of Extracellular Samples

Cell-free supernatants were obtained from approximately 15 mL of overnight cultures of the *Lactobacillus* strains in MRS broth at 30 °C. Then, the cells were separated by centrifugation at 7000× *g*, for 20 min at 4 °C and supernatant was taken and filtered through a cellulose acetate membrane filter (0.2 µm pore size) to eliminate residual bacterial cells. Cell-free supernatant was distributed in 1 mL aliquots, then dH$_2$O (10 mL) and internal standard (0.2 µmol of 10 mmol solution of 2,3,3,3-d4 D,L-alanine) were added to each sample. Three replicate samples were prepared for each *Lactobacillus* strain cell-free supernatant. Un-inoculated MRS broth samples were collected as control. An aliquot of 100 µL of each sample was stored at −80 °C.

## 2.3. In Vitro Antifungal Activity of Lactobacillus

Antifungal activity of *Lactobacillus* strains against *Alternaria alternata* (UNICAM2), *Cladosporium herbarum* (UNICAM10), *Paecilomyces variotii* (UNICAM17), and *Penicillium chrysogenum* (ATCC 9179) indicator strains was assessed in vitro using the agar plate method previously described [4]. The inhibitory activity against the fungal strains *Aspergillus flavus* (ATCC 46283), *Fusarium oxysporum* (UNICAM12), and *Mucor recurvus* (UNICAM15), which develop faster than lactobacilli, was assessed using the dual-culture overlay assay [4]. For all assays, the antifungal activity of each strain was determined by measuring the width of the halo around the bacterial streaks, according to the following 4 steps in a semiquantitative scale: (1) ≥8 mm, (2) 5–7 mm, (3) 3–4 mm, and (4) <3 mm.

## 2.4. Sample Preparation for GC-MS Analysis

Samples were thawed on ice and subsequently 375 µL of a methanol and chloroform mixture (2/1, *v/v*) was added. After 1 h, 380 µL of chloroform and 90 µL of aqueous KCl 0.2 M were added. Samples were vortexed for 10 s and centrifuged at 15,294× *g* for 10 min at 4 °C. A volume of 200 µL of the aqueous fraction was taken and transferred into 1.5 mL sterile glass vials with 10 µL of a 80 mg/L of 2,2,3,3-d4-succinic acid solution. The aqueous fraction was then dried by a nitrogen stream, and derivatized with 40 µL of MSTFA (N-Trimethylsilyl-N-methyl trifluoroacetamide). Samples were kept for 30 min at 70 °C, before adding 600 µL of hexane and then vortexed.

## 2.5. GC-MS Analysis

Samples were analyzed with a Hewlett Packard 6850 Gas Chromatograph, 5973 mass selective detector, and 7683B series injector (Agilent Technologies, Palo Alto, CA, USA) with helium as carrier gas at a flow of 1.0 mL/min. One microliter of each sample was injected with 1 min of split flow delay and resolved on a 30 m × 0.25 mm × 0.25 µm DB-5MS column (Agilent Technologies, Palo Alto, CA, USA). Inlet, interface, and ion source temperatures were 250, 250 and 230 °C, respectively. Oven starting and final temperatures were set at 50 and 230 °C, respectively, with a rate of 5 °C/min for 36 min and then for 2 min at a constant temperature. Electron impact mass spectra were recorded from *m/z* 50 to 550 at 70 eV. Metabolite annotation was achieved by mass spectra comparison with analytical standards, in house library and the NIST14 database (National Institute of Standards and Technology, Gaithersburg, MD, USA).

## 2.6. Multivariate Statistical Data Analysis

As input for multivariate statistical data analysis (MVA), a 27 × 59 matrix, composed of samples × the intensities of the most abundant fragment ion for each metabolite, was constructed. MVA were performed using the software SIMCA-P+ (version 14.1, Umetrics, Umeå, Sweden). GC-MS variables were mean centered and scaled to unit variance. The principal component analysis (PCA) was carried out to study the sample distributions and presence of outliers. For classification of samples to the three classes of *Lactobacillus*, and to find discriminant metabolites, the supervised partial least squares-discriminant analysis (PLS-DA) was used. Importance of the variables in PLS-DA was assessed by considering the

variable importance in the projection (VIP) values. Results of PLS-DA are shown as scatter plot of scores and loadings in the first two principal components, and as a list of metabolites having VIP > 1 attributed to each class of samples on the basis of their coefficients. In the Coefficients Overview Plot the X-variable coefficients, together with their standard errors in cross-validation, for the Y-class are shown. The supervised partial least squares (PLS) regression was applied for the study of linear relationships between antifungal activity and metabolite profiles. Antifungal activity scores were organized in the Y matrix. Results are shown as PLS loading weights plot in first and second components that displays the relation between the X-variables (GC-MS data) and the Y-variables (antifungal activity). Metabolites positively correlated with antifungal high activity lie next to the Y-variable, metabolites negatively correlated lie in the opposite side of the plot. The quality of the PLS and PLS-DA models and the proper number of principal components were assessed on the basis of the cumulative parameters $R^2Y$ and $Q^2Y$ (prediction power calculated in cross-validation), and of their difference value, with a <0.50 threshold.

## 3. Results

### 3.1. GC-MS Metabolite Profiles of Lactobacillus Species

The hydrophilic extracts of samples were analyzed by GC-MS and 59 low molecular weight metabolites were annotated and reported in Table S2. Short-chain carboxylic acids (such as lactic acid, citric acid, succinic acid, and phenyllactic acid), amino acids (valine, leucine, isoleucine, threonine, and pyroglutamic acid), mono- and disaccharides, and polyols were annotated together with fatty acids and analogues, such as oleamide. Analysis of MRS broth chromatograms indicates that it was mainly composed by mono- and disaccharides, citric acid, phosphate, and amino acids essential for growth (results not shown).

To assess differences among the three species of *Lactobacillus*, a PLS-DA was carried out. The discriminant analysis, with a high degree of confidence, well classified samples (score plot in Figure 1a), indicating that the 3 *Lactobacillus* species showed overall different metabolite profiles. By analysis of the loading plot (Figure 1b) and metabolite VIP scores reported in Table 2 (coefficients of metabolites are shown in Figure S1), the fermentation medium of *L plantarum* was found to contain more lactic acid, 3-hydroxy butyric acid (BHBA), 2-hydroxy isovaleric acid (AHVA), 2-hydroxy isocaproic acid (AHCA), succinic acid, 3-phenyllactic acid, and malic acid, together with lower levels of mono- and disaccharides than *L. brevis* and *L. sakei*. In contrast, *L. sakei* showed more sugars. When compared to the other two *Lactobacillus* species, *L. brevis* had more monosaccharides, AHVA, and 4-gamma-aminobutyric acid (GABA) and *L. sakei* showed higher levels of disaccharides. *L. plantarum* and *L. sakei* showed higher levels of pyroglutamic acid. The higher level of saccharides in *L. sakei* and *L. brevis* than in *L. plantarum* was confirmed by the measured carbohydrates fermentation profiles reported in Table S1, where the *L. plantarum* strains, compared to the other 2 species, showed the better fermentation activity.

Broth sugar consumption from *Lactobacillus* strains and production of organic acids are confirmed by Pearson correlation values between GC-MS data depicted as a heat map in Figure S2. In this map, it is clearly visible that organic acids were strongly positively correlated with themselves (r > 0.75) and negatively correlated (r < 0.75) with saccharides (the linear strong negative relationship r = −0.97 between glucose and lactic acid is clearly visible in the Figure S3).

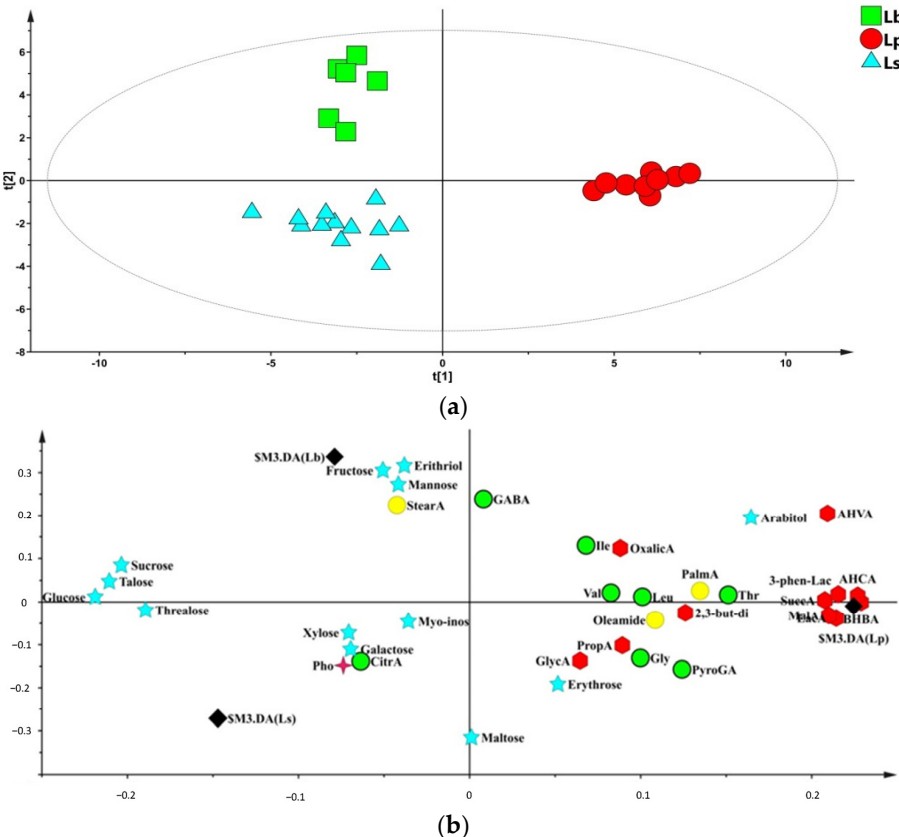

**Figure 1.** PLS-DA of GC-MS data, $R^2Y = 0.93$ and $Q^2Y = 0.88$, over 2 validated components. (**a**) score plot (*Lb*, *Lp*, and *Ls* = *L. brevis*, *L. plantaris*, and *L. sakei*, respectively); (**b**) loading plot, red hexagons = organic acids; green circles = amino acids; light blue stars = saccharides and polyols; yellow circles = fatty acids and analogues; black diamond = loadings of *Lactobacillus* classes. Unknown metabolites are not displayed. Metabolites are abbreviated as in Table S2.

**Table 2.** PLS-DA VIP [a] scores of discriminant GC-MS metabolites.

| *L. plantarum* [b] | | | *L. brevis* | | | *L. sakei* | | |
|---|---|---|---|---|---|---|---|---|
| Metabolite | Class [c] | VIP | Metabolite | Class | VIP | Metabolite | Class | VIP |
| 3-hydroxy butyric acid (BHBA) | OA | 1.25 | Fructose | S | 1.66 | Maltose | S | 1.65 |
| 2-hydroxy isovaleric acid (AHVA) | OA | 1.51 | Erithritol | S | 1.64 | Sucrose | S | 1.23 |
| Arabitol | S | 1.32 | 2-hydroxy isovaleric acid (AHVA) | OA | 1.51 | Glucose | S | 1.20 |
| 2-hydroxy isocaproic acid (AHCA) | OA | 1.24 | Mannose | S | 1.45 | Talose | S | 1.20 |
| Lactic Acid | OA | 1.20 | Arabitol | S | 1.32 | Pyroglutamic acid | AA | 1.09 |
| 3-phenyllactic acid | OA | 1.18 | 4-aminobutyric acid (GABA) | AA | 1.24 | Erythrose | S | 1.04 |
| Malic acid | OA | 1.17 | Sucrose | S | 1.23 | Threalose | S | 1.03 |
| Succinic acid | OA | 1.13 | Stearic acid | FA | 1.21 | | | |
| Pyroglutamic acid | AA | 1.09 | Talose | S | 1.20 | | | |
| Erythrose | S | 1.04 | Threalose | S | 1.03 | | | |

[a] Variable importance in the projection, $R^2Y = 0.93$ and $Q^2Y = 0.88$ over 2 validated components; [b] *Lactobacillus* species with higher level of the metabolite, as indicated by the coefficients; [c] OA = organic acids, AA = amino acids and analogues, S = saccharides and polyols, FA = fatty acids and analogues.

### 3.2. Metabolite Profiles of Lactobacillus Strains

A visual analysis of chromatograms suggests that all the strains consumed sugars and produced lactic acid, though to a different extent, and synthetized oleamide. *L. plantarum* 1/14537 and *L. brevis* DSM 32516 consumed almost all the citric acid of broth origin (data not shown). To study the different production/consumption of metabolites by the nine *Lactobacillus* strains, metabolites indicated by the PLS-DA as the most discriminant between the three species, together with oleamide, were individually measured and results reported as column plot with means and standard deviations (Figure 2). Phenyllactic acid, AHCA, BHBA, AHVA, and malic acid were produced at a greater extent by *L. plantarum*. Pyroglutamic acid was mainly produced by all the *L. plantarum* strains and *L. sakei* S3/1. *L. brevis* M8/1 produced the greatest, by far, quantity of GABA, confirming that different *Lactobacillus* strains within the same species can have their own individual metabolism.

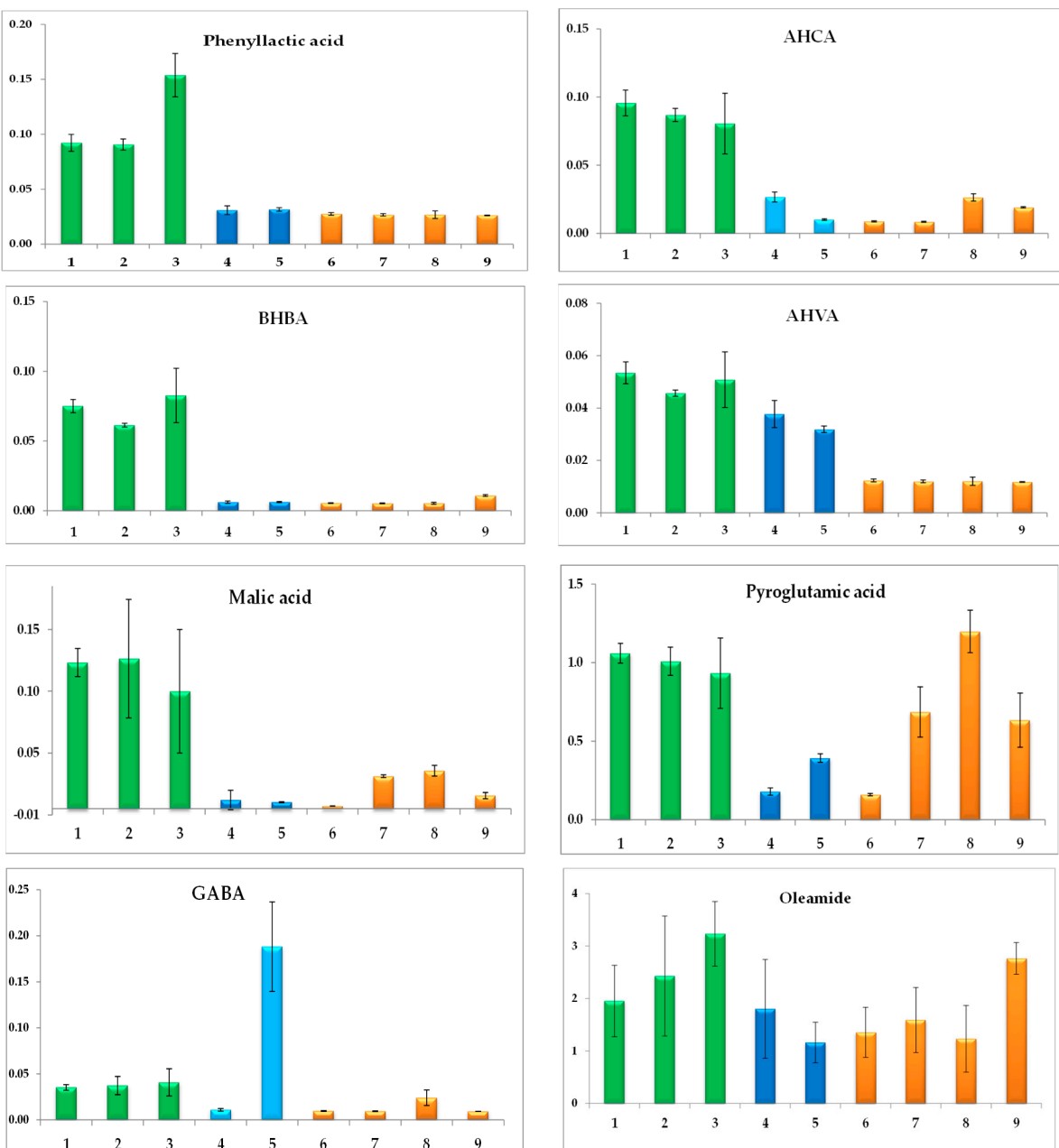

**Figure 2.** Relative abundance of metabolites in each strain: green *L. plantarum* (1 = 1/14537, 2 = 4/16898, 3 = C1/15); light blue *L. brevis* (4 = DSM 32516, 5 = M8/1 S4); yellow *L. sakei* (6 = S4, 7 = S3, 8 = S3/1, 9 = S5). Y-axis values are in A.U. and refer to the intensity of a selected *m/z* ion fragment. Abbreviation of metabolites as in Table S2.

### 3.3. Antifungal Activity and Correlations with Metabolite Profiles

Antifungal activity of *Lactobacillus* strains was also studied. Seven mold species were chosen for their widespread presence in cheese spoilage and for their ability to produce mycotoxins [4]. As shown in Figure 3, the strains exhibited a wide range of antifungal activity, dependent on both fungal species and *Lactobacillus* strain. All *L. plantarum* strains and *L. brevis* DSM 32516 showed the strongest activity with inhibition zones greater than 8 mm against all the fungal strains. The *L. sakei* strains were the less active, with S4 strain showing inhibition zones less than 3 mm against *M. recurvus*, *P. variotii*, and *P. chrysogenum*. *A. alternata* and *C. herbarum* growth was strongly affected (inhibition zone higher than 8 mm) by all the *Lactobacillus* strains except for *L. sakei* S5 which had an inhibition zone lower than 8 mm.

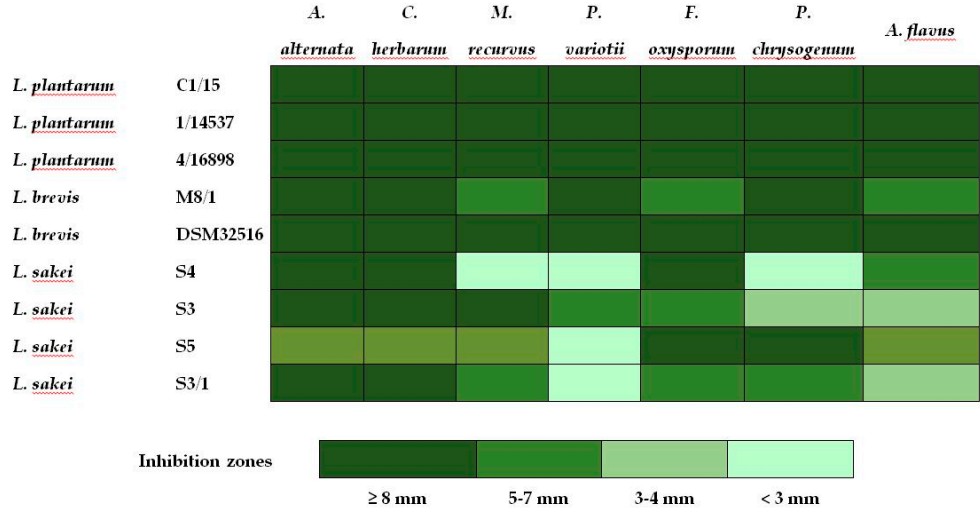

**Figure 3.** Antifungal activity of *Lactobacillus* strains: inhibition zones against *A. alternata*, *C. herbarum*, *M. recurvus*, *P. variotii*, *F. oxysporum*, *A. flavus* ATCC 46283, *P. chrysogenum* ATCC 9179.

These results are in agreement with a previous study [4] where single *L. plantarum* C1/15 strain or multiple *Lactobacillus* strains (*L. plantarum* 4/16898, 1/14537 and *L. brevis* DSM 32516) were able to significantly inhibit the growth of *P. chrysogenum* and *A. flavus* when used as adjunct cultures in the production of miniature Caciotta cheese. With respect to the single *L. plantarum* strain (C1/15), the multiple *Lactobacillus* strain combination resulted more active against the fungal strains tested confirming the synergistic action of different antimicrobial compounds highlighted in the current study [4]. To correlate the metabolite profile of strains to the antifungal activity a PLS regression of GC-MS data was carried out considering the extent of the antifungal activity as Y variable. The resulting loading plot in the first two principal components with superimposed scores of antifungal activity is shown in Figure 4.

In this plot, metabolites that lie next to the antifungal activity have positive relationship with this latter, metabolites in the opposite side of the plot, passing from the origin are inversely related with the activity. On the basis of these rules, we can note that organic acids and arabitol were directly involved in the activity against *P. chrysogenum*, *P. variotii*, and *M. recurvus*. Oleamide was correlated with *A. flavus* and *F. oxysporum*.

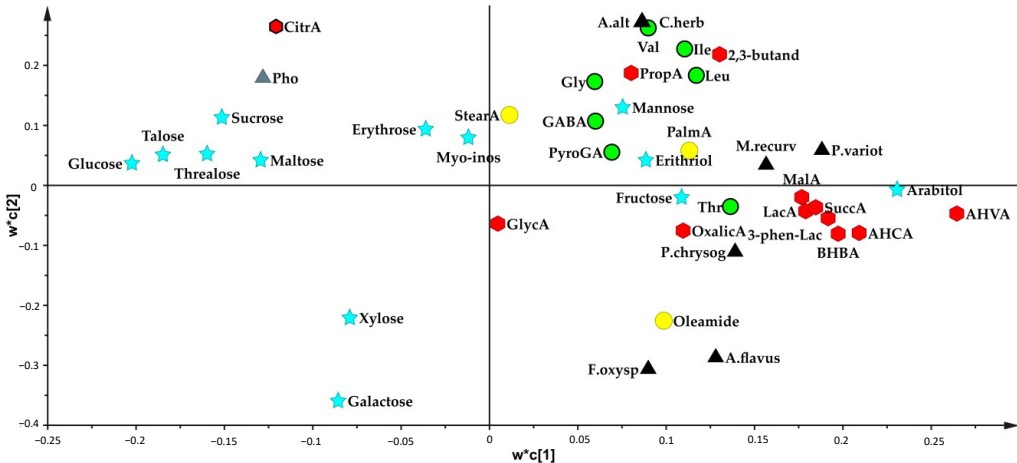

**Figure 4.** PLS variable loadings plot of GC-MS data (X-variables) superimposed with antifungal activity (Y-variables). $R^2Y = 0.89$ and $Q^2Y = 0.65$ over 5 validated components. Black triangles = Y coefficients for antifungal activity; red hexagons = organic acids; green circles = amino acids; light blue stars = saccharides and polyols; yellow circles = fatty acids and analogues; black diamond = loadings of *Lactobacillus* classes. Unknown metabolites are not displayed. Metabolites are abbreviated as in Table S2.

## 4. Discussion

The inherent ability of *Lactobacillus* to produce compounds antagonistic to food microbial contaminants as a part of their natural defense mechanism can provide the dairy food industry with natural and healthier alternatives to chemical preservatives [15], especially for fresh products. The antifungal activity of *Lactobacillus* has been related to the synergistic effect of several compounds, and part of these compounds shows nutraceutical properties.

Compared to the other species here studied, *L. plantarum* strains showed more organic acids, among which 3-phenyllactic acid, which shows a broad spectrum of antimicrobial properties, is effective against bacteria and fungi including yeast [16]. In sourdough bread started with *L. plantarum* 21B, the onset of growth of *Aspergillus niger* was delayed 7 days, with respect to bread started with a *L. brevis* strain not producing phenyllactic acid [17]. However, it has been reported that phenyllactic acid is active against fungal species only at mg mL$^{-1}$ concentrations, suggesting an overall antifungal effect in association with other factors produced by LAB [17]. 3-Phenyllactic acid is a by-product of phenylalanine metabolism in *Lactobacillus*, where phenylalanine is transaminated to phenylpyruvic acid and further reduced to 3-phenyllactic acid by 2-hydroxy dehydrogenase [15]. GABA is another product of amino acid metabolism, biosynthesized by microorganisms through decarboxylation of glutamate by glutamate decarboxylase [18]. GABA is an important bioactive compound and its production by various microorganisms has been actively explored [18]. Higher levels of GABA were detected solely in the *L. brevis* M8/1 strain. Besides lactic acid, a number of 2-hydroxy organic acids, that show antimicrobial properties [10], are generally produced by *Lactobacillus*. Organic acids levels were found correlated with the antifungal activity against *P. chrysogenum*, *P. variotii*, and *M. recurves*. *L. plantarum* was found to produce hydroxy organic acids, some of which (AHVA and AHCA) have shown antifungal proprieties [10]. In particular, AHCA has been studied for its potential in inhibiting cell growth and biofilm formation of the pathogenetic *Aspergillus* and *Candida* species in topical use [19,20]. One of the most important virulence factors of *C. albicans* is the ability to form biofilms that protect this yeast against endogenous and exogenous inhibitory substances, in fact almost 65% of microbial infections in humans are biofilm-related [19]. The effects of oral administration of lactobacilli against biofilm-associated infections have been studied [21]. All the strains synthetized oleamide, not present in the broth, and levels of this compounds have been correlated with the antifungal activity against *A. flavus* and *F. oxysporum*. Oleamide in an amide of oleic acid present in small amount in animal brains and can be produced by microorganisms [22]. In dairy products'

fermentation processes, oleamide is synthesized from oleic acid, which is an abundant component in these food products, owing to lipase enzymatic amidation [23]. Epidemiological and clinical data have shown that fermented dairy products possess preventive effects against dementia, including Alzheimer's disease, and those effects have been linked to oleamide [23]. This compound has been identified as the agent responsible for reducing microglial inflammatory responses and neurotoxicity [23]. Oleamide is also an endogenous bioactive signaling molecule that acts in various cell types and could elicit different biological effects [23]. Among the wide range of functions, the most acknowledged one is its sleep-inducing effect [24]. It has been found that *L. plantarum* 1/14537 and *L. brevis* DSM 32516 consumed almost all the citric acid of broth origin. Citric acid fermentation by LAB could lead to acetoin and diacetyl production which are aromatic compounds. However, obligately heterofermentative lactobacilli such as *L. brevis* when present at high levels and facultatively heterofermentative lactobacilli such as *L. plantarum*, under certain conditions, by co-metabolization of citrate and different sugars may produce great amount of $CO_2$, giving rise to holes, splits and blowing defects of aged cheeses [25]. *L. brevis* strains are also characterized by other biochemical properties such as production of $CO_2$ from glucose and ammonia from arginine, as shown in Table S1, that are generally used as tool for lactobacilli classification [26]. Moreover, *L. brevis* could contribute to the degradation of arginine in cheese that can lead to ornithine accumulation by the arginine-urease or the arginine deaminase pathways [27]. On the other hand, ornithine and other free amino acids could be decarboxylated to putrescine and other biogenic amines that in high concentration could be responsible for toxic effect on human health.

In conclusion, results of this study provide reference data for interpreting the differences in the metabolite profiles of *Lactobacillus* strains isolated from different fermented food products and their correlation with antifungal activity in vitro. These autochthonous strains could be considered potential good candidates to be used in cheese manufacturing as bioprotective cultures and for in situ production of postbiotic substances. However, further studies in cheese model systems are warranted to evaluate in situ positive metabolic activities such as organic acids production, GABA and oleamide formation, as well as negative activities including biogenic amines production and the potential to cause blowing defects in the initial and later stages of ripening.

**Supplementary Materials:** The following are available online at https://www.mdpi.com/article/10.3390/dairy2030026/s1, Figure S1: PLS-DA coefficient overview plot; Figure S2: Heat map of Pearson correlation values of GC-MS data matrix; Figure S3: Correlation plot of lactic acid (X-axis) vs. glucose (Y-axis) levels. Table S1: Carbohydrates fermentation profiles of Lactobacillus strains; Table S2: List of GC-MS metabolites.

**Author Contributions:** Conceptualization, P.C., M.B.P., S.C., and P.S.; Methodology, P.S., A.M., and M.B.P.; Software, P.S.; Validation, P.S. and M.B.P.; Formal Analysis, P.S., A.M., and M.B.P.; Writing—Original Draft Preparation, P.S., P.C. and M.B.P.; Writing—Review and Editing, P.S., P.C., S.C. and M.B.P. All authors have read and agreed to the published version of the manuscript.

**Funding:** This research received no external funding.

**Informed Consent Statement:** Not applicable.

**Data Availability Statement:** Not report data available.

**Conflicts of Interest:** The authors declare no conflict of interest.

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
