# Peer review of "GC-MS Metabolomics and Antifungal Characteristics of Autochthonous Lactobacillus Strains"

_2624-862X, doi:10.3390/dairy2030026_

Round 1

Reviewer 1 Report

No hypotheses were stated in Introduction and tested in Conslusions

Reviewer 2 Report

Results suggest that GC-MS based metabolomic analysis is a useful tool for future investigations on the characterization of Lactobacillus strains as biopreservatives.....

In general, the introduction of new analytical techniques that will improve the quality of research, accelerate the obtaining of results is innovative.
I work on GC myself but with FID and many people are afraid to introduce new methodologies for GC determination.

Works that show new methodologies are perfect for me, because apart from the field of animal husbandry, I am also a chemist.

The study is relevant and interesting.

The paper is well written.
The text is clear and easy to read.
The conclusions are consistent with the evidence and arguments presented.
The authors address the main question posed.

the only mistake is the wrongly quoted literature
Materials and Methods - in italics

Set as a separate point
In conclusion, results of this study provide reference data for interpreting the differences in the metabolite profiles of Lactobacillus strains isolated from different food sources and their correlation with antifungal activity.

Reviewer 3 Report

All the comments are listed in the manuscript.

Reviewer 4 Report

The authors evaluated the in vitro metabolomic profile by GC-MS and antifungal activity from various lactobacilli as potential candidates as bioprotective cultures for cheeses. This is an interesting study, since the use of selected microorganisms may have a positive impact on dairy processing industry, but I would suggest some improvements:

Line 17 and 19: The authors indicated “richer in organic acids”, “higher levels of sugars” and “higher correlations”, but the sentences are not clear. When referring to these types of terms, authors should compared with other treatments (richer or higher than XXX). The word “richer” is not appropriate for scientific language and should be replaced along the manuscript (also in lines 149, 153).

Line 19: Authors should indicate level of correlation and its significance (P-value).

Line 32: Replace researches for researchers

Line 44: “thanks” does not have scientific soundness. Authors should replace it for “due”.

Line 52: After reference 4, Authors should emphasize the limitation of the previous study (did not know the compounds with antifungal activity) and should include a research question/hypothesis that would be in accordance with the objectives of the study.

Line 65-73: Only title of section 2.1 should be in italics.

Line 69: Please use proper sign of degree (°)

Line 72: replace assed by assessed

Line 78: g should be in italics.

Line 96: “an aliquot of 100 ul”. Authors should clarify this, since in Section 2.2 samples were freeze dried.

Line 97: g force should be in italics. In addition, centrifugation temperature should be reported.

Line 98: What is methanol/water fraction? According to the methodology, samples were mixed with chloroform. Please clarify.

Line 116: Please describe what MVA is.

Line 118. Principal component analysis should not be in capital letters.

Line 122: variable importance in the projection should not be in capital letters.

Line 152: Authors should replace “As opposite” for “In contrast”.

Line 164: authors should replace L. for lactobacilli.

Line 169 and 208: “Here” does not have scientific soundness.

Line 169: r value range and level of significance should be indicated.

Line 175: fermentation of citric acid could be a concern in cheeses, since it may lead to the formation of other defects, such as gas. As also shown in Table S1, authors should address the occurrence of other potential defects associated with fermentation of glucose (into CO2) and decarboxylation of arginine (into biogenic amines). Those lactobacilli are not necessarily good candidates.

Lines 178-182: Sentences are confusing and should be in accordance with the order of graphs shown in Fiure 2.

Line 213: In the discussion, were these results in accordance to what was found in the previous study (reference 4)?.

Lines 254-256: Authors should emphasize that these findings were made under in vitro conditions and further work might be necessary to evaluate on a cheese matrix.

Line 268: references 3, 6, 11, 13, 15, 17, 18, 19 and 21 should have a proper DOI format (as shown in other references). Reference number 20 should have the name of the journal in capital letters.

Round 2

Reviewer 3 Report

The authors made all the corrections.

Reviewer 4 Report

The authors made a great effort on improving the manuscript. I accept it after only minor observations, which are detailed below:

-Line 54-56: Merge the objective parragraph with the one indicated in lines 61-63.

Line 105: What MSTFA is?

-Line 148: authors should include after parragraph of MRS composition"(results not shown)"

-Line 187: Authors should replace "peculiar" with "individual".

-Line 204: Authors should replace: "in line" with "in agreement".
